# FINDING BETTER PROTOTYPES FOR INTERPRETABLE TEXT CLASSIFIERS WITH LLM OPTIMIZATION

## ABSTRACT

Prototype neural networks are the most popular form of interpretable-by-design classifiers in machine learning. Within this field, prototypes are typically learned as black-box vectors, and then projected onto the nearest example from the training data for visualization and inference purposes. This improves interpretability because we can understand the logic behind predictions based on the similarity between the input instance and the nearest prototype in the network. However, because these prototypes are real training instances there are at least two major issues with this approach. Firstly, as the projected prototypes do not represent the learned "black-box" vectors which were optimized for accuracy, there is typically a performance drop off. Secondly, because the prototypes are real training instances, they are usually overly specific and full of spurious or irrelevant details, making them difficult to interpret readily. In this study, we address this problem by using large-language models (LLMs) as a tool for optimization to find better prototypes for the network. Across a series of experiments, we find that our method produces prototypes which sacrifice less performance and are more intelligible compared to baselines which project. Previously, it was not possible to visualize a learned prototype, because methods were constrained to projection using actual training data, but our approach suggests a possible path to overcome this limitation.

## 1 INTRODUCTION

The problem of interpretability in machine learning (ML) is perhaps the core issue preventing widespread trust of the technology globally (Amodei, 2024). If we are ever to actually appropriately trust ML in sensitive domains, more fundamental progress is urgently needed to address issues of alignment, safety, and reliability (Sharkey et al., 2025). Out of this urgency, researchers have proposed various solutions roughly divided into post-hoc justifications for predictions (Ribeiro et al., 2016), and interpretable-by-design architectures (Chen et al., 2019). In the latter category, prototype neural networks are the most popular architectural design methodology (Li et al., 2018), as they have strong justification and roots in psychological theory (Rosch, 1973), as well as great practical benefit in ML (Chen et al., 2023b; Cunningham et al., 2003; Kenny et al., 2023). However, these methods have struggled with perhaps the most fundamental problem in their design methodology, which is how to actually visualize the learned prototypes the network produces.

Historically, researchers have approached the problem by learning a set of black-box prototype vectors in the penultimate layer of the neural network, and then once training is complete, projecting these onto the closest real examples in the training set (Chen et al., 2019; Ming et al., 2019; Ma et al., 2024). Although this approach has proven widely successful, it poses at least two severe problems. Firstly, although not always, this approach has historically resulted in a drop in model performance, which initially hampered their adoption (Chen et al., 2019; Ma et al., 2024), because training examples cannot realistically align with the "perfect" learned black-box prototype. Secondly, and perhaps more importantly, this clashes with classical prototype theory outlined by Rosch (1973), which states that prototypes should be a more abstract representation of a class, which generalizes across instances and is not overly specific. However, especially in textual domains, as the training data are often long sequences of text with many irrelevant features, this is often not possible to achieve. To address these issues, we propose to use large-language models (LLMs) as an optimization tool to find textual inputs that not only map better to the learned prototypes, but also better align with prototype theory by being sparse featural representations of the class, as we illustrate in Figure 1.

**Projected Prototype (Nearest Neighbour from Training Data)**

Sloppily directed, witless comedy that supposedly spoofs the "classic" 50s "alien invasion" films, but really is no better than them, except of course in the purely technical department (good makeup effects). And any spoof that is worse than its target is doomed to fail ("Casino Royale", "Our Man Flint" are worse than almost any James Bond movie). After two hours of hearing the screeching voices of the aliens, you'll be begging for some peace and quiet. (*1/2)

**LLM Optimized Prototype (Generated by LLM)**

Spoof movie with decent technical quality, failing to deliver laughs.

Figure 1: Our Proposal: Typically in the prototype literature researchers have projected prototypes onto the nearest training example to visualize them and run inference. However, this can result in overly long, complex, and cumbersome prototypes. In contrast, we propose to begin an optimization process from the nearest neighours pool, which uses LLMs as optimization tools to iteratively refine a simpler, more intelligible version of the prototype, which often better aligns with the learned prototype vector. In this real example from our tests, the projected prototype discusses overly specific features such as *"Casino Royal"* and *"James Bond"*, but our optimized prototype abstracts this down to a high-level set of features involving spoof movies, technical quality, and humor, which are much more general and emblematic of prototype theory (Rosch, 1973).

In Section 2, we outline prior work in the area for context. Section 3 describes our architecture for training interpretable-by-design textual classifier neural networks, and our key innovation involving the usage of LLMs as an optimization tool to discover the learned prototypes. Section 4 runs our algorithm on six benchmark datasets across three models, demonstrating that intelligibility and accuracy are better achieved with our method compared to baseline approaches before a final discussion in Section 5.

To summarise, our three main contributions are as follows:

- We introduce a new paradigm for how to train interpretable-by-design prototype neural networks using LLMs as an optimization tool to try and discover the learned prototypes, rather than simply projecting them onto the nearest training instances.

- Our experiments show that the final prototypes produced by our optimization algorithm can radically simplify prototypes down to their core concepts and in doing so not compromise accuracy.

- We demonstrate this algorithm can also increase the accuracy of prototype neural networks compared to projection-based baselines when significant performance is lost due to prototype projection.

## 2 RELATED WORK

Prototype theory was first proposed by Rosch (1973) in the 1970s, the hypothesis was that humans learn abstract representations of a class for classification purposes, and indeed it was one of the first ideas from psychology imported into ML (Kim et al., 2014), largely inspired by earlier work in case-based reasoning (Smyth et al., 2001). In this section, we review initial work in the area, before a more specific discussion about text-based networks and the usage of LLMs as optimization tools.

### 2.1 INITIAL WORK

Prototype networks were first inspired by work in case-based reasoning (CBR) (Cunningham et al., 2003), which is based on the psychological theory that humans reason about new states based on their past history (i.e., cases), and use these past experiences to make decisions in the present. Kim et al. (2014) did foundational work in the area with their Bayesian case model, and this was followed with related CBR work in various guises across domains (Papernot & McDaniel, 2018; Kenny et al., 2023).

However, it was Chen et al. (2019) who inspired a flurry of work in prototype networks, and helped popularize them into the prominent interpretable-by-design architectural choice (Ma et al., 2024).

## 2.2 Text-based prototype networks

Inspired by the work in images, text-based networks began to emerge. First was the work by Ming et al. (2019), who outlined a general framework for applying the work of Li et al. (2018) to text domains. The authors demonstrated widespread utility of the framework, and we adapt it here for our computational evaluation later. Van Aken et al. (2022), used prototype networks for medical classification; departing from prototype projection onto nearest training examples, the authors leave the learned "black box" prototype vectors as is to preserve classification performance, and simply show the nearest training examples without prototype projection. This is similar to Das et al. (2022) who proposed ProtoTex, and innovated with counterfactual 'negative' prototypes, but again opted to avoid prototype projection. This practice of not projecting prototypes has been criticized lately in the literature by Ma et al. (2024), who point out this practice does not result in interpretable networks, and is just done to avoid drops in classification performance. Having said this, there has still been a wave of work adopting the approach of prototype projection, either on the case level (Wen et al., 2025; Wen & Rezapour, 2025), the sentence level (Hong et al., 2023; Wei & Zhu, 2024), or just general evaluation of the frameworks (Sourati et al., 2024; Hanawa et al.; Davoodi et al., 2023). We offer a fresh perspective on this long standing problem, instead of projecting prototypes onto training examples (and often losing performance), or leaving them as black box vectors (and losing interpretability), we attempt to discover the actual learned "black-box" prototype vectors by optimizing towards them with LLMs.

## 2.3 Language Models as Optimizers

Since the recent success of generative AI, researchers have tried to capitalize upon LLMs as optimization tools to solve previously unapproachable problems. Early work in this area mainly worked by finetuning LLMs to be optimizers (Meyerson et al., 2023; Lehman et al., 2023; Chen et al., 2023a). However, recently Yang et al. (2024) looked at just using prompting itself to solve toy problems such as linear regression and the traveling salesman problem, and had reasonable success in doing so. We are inspired by this idea, but interested in the problem of using LLMs to visualize learned prototypes for more interpretable and accurate text classifiers.

## 3 Method

In this section we describe our backbone architecture for training these networks, which is a generalization over prior work (Ming et al., 2019; Das et al., 2022; Ma et al., 2024). We assume some encoder $f(\cdot)$ comprising of frozen parameters $f_{enc}(\cdot)$, and a subsequent set of trainable ones $\phi(\cdot)$. In our setup, converging with the empirical observations of Sourati et al. (2024) for best performance, $f_{enc}(\cdot)$ represents all blocks of our text classification models, except for the final trainable block $\phi(\cdot)$. Thus, considering some datum $x_i \in \{X\}_{i=0}^N$, its final latent representation may be obtained via the transformation $z = \phi(f_{enc}(x_i))$. In a forward pass through the network, a datum $x_i$ has its similarity measured against $p \in \{P\}^m$ prototypes to form an activation vector $a_i \in R^m$. Finally, $a_i$ is fed to a linear weight matrix $W \in R^{(m,c)}$ to give the final classification logits, where $c$ is the number of classes.

**Similarity Function.** Following more recent trends in prototype-based networks (Ma et al., 2024), and mirroring common practice in NLP (Mikolov et al., 2013), we use cosine similarity as our similarity function:

$$\cos(z_i, P_k) = \frac{z_i}{\|z_i\|_2} \cdot \frac{P_k}{\|P_k\|_2} \tag{1}$$

Although largely an insignificant hyperparameter (Sourati et al., 2024), this choice of similarity function ensures the focus on angular similarity between representations. Empirically, we also found it supported better convergence than $L_2$ norm-based distance functions which are another popular choice (Chen et al., 2019; Das et al., 2022; Kenny et al., 2023).

## 3.1 LOSS FUNCTION

Following the majority of work in this field (Ming et al., 2019; Sourati et al., 2024; Das et al., 2022), our loss function consists of four components that are all minimized during training: a classification loss, interpretability loss, clustering loss, and separation loss.

The classification loss $L_{ce}$ is the standard cross-entropy loss between predicted and true labels that we seek to minimize:

$$L_{ce} = -\sum_{i=1}^{N} \mathbf{y}_i \cdot \log(\text{SoftMax}(W^T \cdot a_i)) \tag{2}$$

where $a_i = [\cos(z_i, p_1), \cos(z_i, p_2), \dots, \cos(z_i, p_m)]$ is the activation vector of cosine similarities between example $x_i$ and all prototypes, $\mathbf{y}_i$ is the one-hot encoded true label vector for example $x_i$, and $N$ the batch size.

The interpretability loss $L_i$ ensures that prototypes remain interpretable by maximizing the similarity between each prototype and its closest training sample:

$$L_i = \frac{1}{M} \sum_{k=1}^{M} \min_{j \in \{1,\dots,N\}} (1 - \cos(p_k, z_j)) \tag{3}$$

This constraint keeps prototypes close to actual training samples by maximizing their similarity to the nearest training examples, allowing them to be represented by their closest training examples. Subtracting the cosine similarity from 1 allows us to form this as a minimization objective with a lower bound of 0.

The clustering loss $L_c$ ensures that training examples are close to at least one prototype by maximizing the similarity between each training example and its nearest prototype:

$$L_c = \frac{1}{N} \sum_{j=1}^{N} \min_{k \in \{1,\dots,M\}} (1 - \cos(z_j, p_k)) \tag{4}$$

where $p_k$ represents the $k$-th prototype, and $z_j$ is the encoded representation of example $x_j$. Minimizing this loss pulls training examples closer to their nearest prototypes.

The separation loss $L_s$ encourages prototype diversity by minimizing the cosine similarity between different prototypes:

$$L_s = \frac{2}{M(M-1)} \sum_{\substack{k,l \in \{1,\dots,M\} \\ k<l}} (1 + \cos(p_k, p_l)) \tag{5}$$

where we sum over unique pairs $(k, l)$ with $k < l$ to avoid double counting. Since cosine similarity $\cos(p_k, p_l)$ ranges from $[-1, 1]$, the term $(1 + \cos(p_k, p_l))$ ranges from $[0, 2]$. Minimizing this term pushes prototypes apart in the embedding space by penalizing high similarity, with the minimum value of 0 achieved when prototypes are maximally dissimilar (cosine similarity = -1).

The total loss is defined as:

$$L = L_{ce} + \lambda_i L_i + \lambda_c L_c + \lambda_s L_s \tag{6}$$

where $\lambda_c, \lambda_i, \lambda_s \geq 0$ are regularization factors that control the contribution of each auxiliary loss term. Mirroring Ming et al. (2019), we set these to $\lambda_i = 0.1$, $\lambda_c = 0.01$, and $\lambda_s = 0.01$. Due to the careful formulation of each term, all components are minimization objectives with a well defined lower bound, allowing more stable training.

## 3.2 LLM OPTIMIZATION

Prototype-based text classifiers learn latent prototype vectors $\mathcal{P} = \{p_1, p_2, \dots, p_m\}$ that, while effective for classification, lack direct interpretability. These learned prototypes exist in high-dimensional embedding spaces and can only be understood through their nearest training examples, which may not capture the essential characteristics that make them effective decision boundaries (Chen et al., 2019). We propose a novel approach that leverages large language models (LLMs) to discover

these abstract prototypes as interpretable, optimized text representations that maintain or improve classification performance.

Formally, given a learned prototype $p_k \in \mathbb{R}^d$ and an LLM $\mathcal{L}$, our objective is to find an optimal textual representation $t_k^*$ such that:

$$t_k^* \approx \arg\max_t \cos(\phi(f_{enc}(t)), p_k) \tag{7}$$

where $t$ is a textual input to the classifier generated by an LLM, and $t_k^*$ should give an approximation of the prototype which is short, focuses on core concepts in the domain, highly general, and minimizes the presence of irrelevant features.

### 3.2.1 INITIALIZATION

Our method begins by generating short, focused textual guesses using the nearest neighbor training examples as a guide. For each learned prototype $p_k \in \mathcal{P}$, we first identify the top-$k$ nearest training examples $\mathcal{NN}_k = \{x_{i_1}, x_{i_2}, \ldots, x_{i_k}\}$ based on cosine similarity in the embedding space. Rather than using these examples in our meta prompt and iteration loop, we first prompt $\mathcal{L}(\cdot)$ to generate concise text snippets $\mathcal{T}_k^{(0)}$ that capture the essential concepts observed in these examples, this minimizes the context window and escapes the local minima around the prototype, which often biases longer text. In practice, we often found this was necessary if similarity to $p_k$ was already high, as it was difficult for $\mathcal{L}(\cdot)$ to guess better examples which improved upon the similarity scores. This resulted in worse initial guesses, but allowed the process to optimize towards the learned prototype using shorter textual examples, ultimately resulting in the same (or better) performance. In the end, this had the effect of speeding up the optimization loop, and finding shorter, more concise prototype

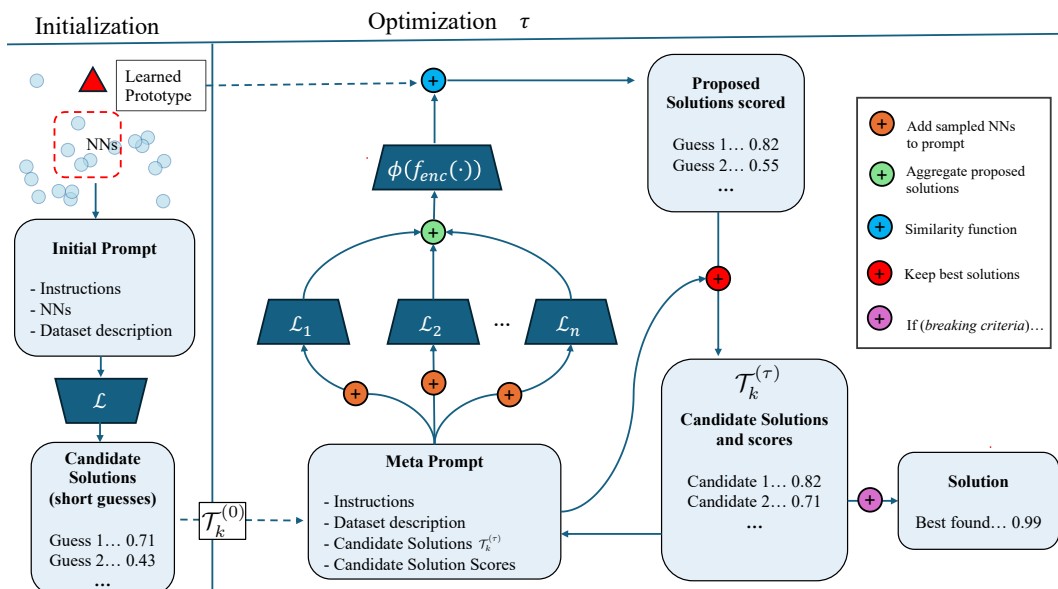

Figure 2: Method overview: Initially we construct a prompt from the closest neighbours to the learned prototype we are trying to visualize and the dataset specific description. This prompt generates a set of $n$ short textual guesses for the prototype which are used to begin the optimization process. Each optimization iteration constructs a meta prompt from the dataset specific instructions, the current candidate solutions, and their respective scores. The meta prompt is duplicated $n$ times, each with randomly sampled nearest neighbors to the prototype to help diversity in the LLM's new guesses. The guesses from each LLM are aggregated and passed into our classifier's encoder to get their representations. These encoded guesses then have their similarity measured to the prototype, and are each scored, if any are better than the current candidate guesses, they replace them. At this point, if the stopped condition is met, the highest scoring guess is used for prototype projection.

texts. We construct an initial prompt $\pi_{init}(\mathcal{NN}_k)$ and extract candidate texts from the LLM response, selecting those with highest similarity to $p_k$ to form our initial candidate set $\mathcal{T}_k^{(0)}$.

### 3.2.2 ITERATIONS AND STOPPING CRITERIA

At each iteration $\tau$, we maintain a set of candidate prototype texts which represent the current best guesses from $\mathcal{L}(\cdot)$ across all iterations with:

$$\mathcal{T}_k^{(\tau)} = \{t_k^{(\tau,1)}, t_k^{(\tau,2)}, \ldots, t_k^{(\tau,|\mathcal{T}_k^{(\tau)}|)}\} \tag{8}$$

for each prototype $p_k$, where $\mathcal{T}_k^{(\tau)}$ denotes the set of candidate prototype texts for prototype $p_k$ at iteration $\tau$, and $t_k^{(\tau,i)}$ represents the $i$-th candidate text in this set. We compute similarity scores for all current candidates using $s_i^{(\tau)} = \cos(\phi(f_{enc}(t_k^{(\tau,i)})), p_k)$ and construct an optimization prompt that includes current candidates with their scores, along with sample training examples from $\mathcal{NN}_k$. The LLM generates new candidate texts, which we evaluate and use to update our candidate set by replacing lower-scoring texts with better alternatives that achieve higher similarity to the target prototype $p_k$ such that:

$$\mathcal{T}_k^{(\tau+1)} = \text{TopK}\left(\mathcal{T}_k^{(\tau)} \cup \mathcal{G}_k^{(\tau+1)}, K\right) \tag{9}$$

where $\mathcal{G}_k^{(\tau+1)}$ represents the new candidate texts generated by the LLMs at iteration $\tau + 1$, and $\text{TopK}(\cdot, K)$ selects the $K$ texts with highest cosine similarity scores $\cos(\phi(f_{enc}(t)), p_k)$.

To maximize exploration at each iteration, we employed a set of LLMs such that $\mathcal{L} = \{\mathcal{L}_1, \mathcal{L}_2, \ldots, \mathcal{L}_n\}$. This set of LLMs process the same meta prompt customized with different sampled training examples from $\mathcal{NN}_k$, serving to ensure each prompt receives variation while focusing on the relevant part of the latent space containing the concept we seek to refine. The stopping criteria can either be when a near-perfect solution is found such that:

$$\max_{t \in \mathcal{T}_k^{(\tau)}} \cos(\phi(f_{enc}(t)), p_k) \approx 1.0, \tag{10}$$

or a set number of iterations is performed. In practice, it is unlikely to map to a perfect solution as the prototype may lay slightly off the textual data manifold, so a threshold would be needed for the prior stopping approach.

### 3.2.3 IMPLEMENTATION DETAILS

We employed 3 parallel LLM instances with `Meta-Llama-3-8B-Instruct`. We set the number of candidate solutions at each iteration to a maximum of 10, meaning the algorithm could replace its current 10 best candidates with potentially better ones from the pool of 30 newly generated texts from 3 LLMs (each generating 10 solutions). When showing random training examples in each prompt at each iteration, we sampled 2 from the nearest neighbor pool $\mathcal{NN}_k$ with $|\mathcal{NN}_k| = 20$. Our stopping criteria was to terminate the optimization after 20/15 iterations on our two set of experiments, respectively.

**Prototype Projection.** After optimization, each learned prototype $p_k$ is projected onto its best textual approximation from the LLM in the embedding space:

$$p_k^{proj} = \phi(f_{enc}(t_k^*)) \quad \text{where} \quad t_k^* = \arg\max_{t \in \mathcal{T}_k^{(T_{max})}} \cos(\phi(f_{enc}(t)), p_k) \tag{11}$$

This projection ensures that each prototype can be directly interpreted through its optimized textual representation $t_k^*$, facilitating human understanding and analysis of the model's decision-making process. In our baseline methods, we project each prototype onto its nearest training example: $p_k^{proj} = z_{j^*}$ where $j^* = \arg\max_{j \in \{1,\ldots,N\}} \cos(p_k, z_j)$. As Theorem 2.1 from ProtoPNet (Chen et al., 2019) states this stage should only result in accuracy performance drop if the similarity difference is too great. We study situations when this gap is small and large in the next section to see how our algorithm can help in both circumstances.

### 3.3 HYPERPARAMETER CHOICES

As LLMs are typically familiar with NLP datasets, we found adding dataset-specific descriptions improved the guesses at initialization. The number of nearest neighbors is dataset specific and a hyperparameter to be tuned, we empirically found via a grid-search that if the number was too high, the information given to the LLM(s) is too volatile and disrupts optimization, likewise we found if the number was too low, then diversity suffers and early local minima become problematic. The number of LLMs used is important as the more one can use, the more guesses that are possible each iteration, and the better convergence is (Yang et al., 2024). We found our LLM was only capable of making 10 guesses reliably each iteration, so by using three in parallel we could triple the amount of guesses, which again helps avoid early local minima (Yang et al., 2024).

## 4 EXPERIMENTS

We were interested in testing two hypothesis. First, we anticipate that our method could be used to simplify overly long, complex prototypes to their core semantic meaning without losing accuracy. Second, it is hypothesized that in cases where prototype projection results in a significant decrease in classifier accuracy, our approach can converge to a better solution which preserves some of the delta loss. The next two sections put these hypotheses to the test.

### 4.1 OPTIMIZING PROTOTYPE INTELLIGIBILITY

In this section we test if our method can simplify complex prototype textual representations into more intelligible ones. To do this, we utilize three popular models in the literature for text-based prototype models (Das et al., 2022), BERT (Devlin et al., 2019), Electra (Clark et al., 2020), and RoBERTa (Liu et al., 2019).[1] All three had their parameters frozen except for the final block $\phi(\cdot)$ which was finetuned to help learn better representations (Sourati et al., 2024). For datasets we consider IMDB (Maas et al., 2011), Amazon Reviews (McAuley & Leskovec, 2013), and AG News (Zhang et al., 2015), because their typical textual datum are suitably long. We set the number of prototypes in each model equal to three times the number of classes to increase prototype variability while maintaining computational tractability, and allowed our algorithm to optimize for 20 iterations per prototype. Additional details are in Appendix A and B. All tests were run across 6 seeds, with mean and standard error reported.

---

[1]In early tests we also considered Llama and Qwen LLM models as backbone encoders, but we found they performed worse so ultimately omitted them from the results.

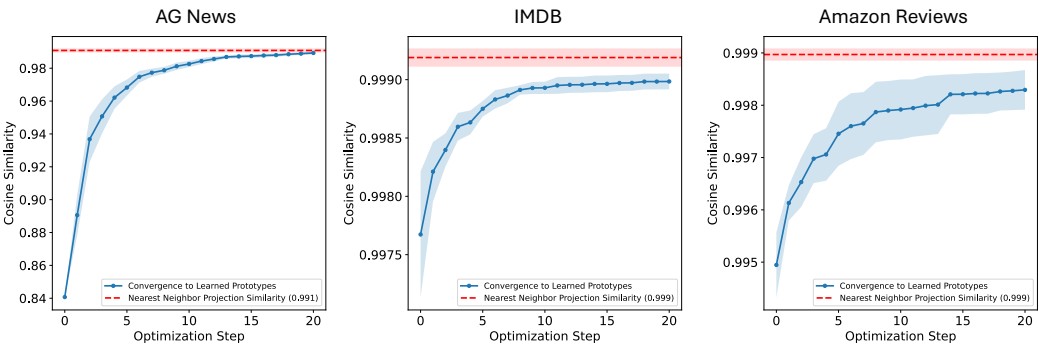

Figure 3: Optimization curves on the intelligibility experiment: All models and datasets converged gradually towards better prototypes which near equaled the original projection prototype's similarity scores, but were on average only 13.9% of the number of characters in length, thus helping to optimize interpretability of the prototypes. AG News had the initial biggest drop in similarity when beginning optimization, but quickly converged nonetheless towards a similar result to the other datasets. All results are averaged across models, and show mean and standard error across 6 random seeds.

Table 1: Intelligibility experiment results: Overall, our LLM-based optimization approach to prototype discovery achieved notably shorter prototypes. Specifically, the average character length of our prototypes was 95.5, compared to using the training data as projections which was 682.3. There was a relatively insignificant drop in accuracy with our approach overall corresponding to 0.02%. Length is represented by the number of characters in the text. The change in accuracy is represented by $\Delta$.

| Dataset | Model | Learned Acc. | Optimized Prototype Acc. | $\Delta$ | Length | Projected Prototypes Acc. | $\Delta$ | Length |
|---|---|---|---|---|---|---|---|---|
| IMDB | MPNet | $92.70 \pm 0.11$ | $92.69 \pm 0.12$ | $-0.01$ | 67.7 | $92.70 \pm 0.10$ | $-0.01$ | 898.1 |
| | Modern-Bert | $87.30 \pm 0.24$ | $87.27 \pm 0.21$ | $-0.03$ | 115.2 | $87.32 \pm 0.23$ | $+0.02$ | 846.7 |
| | Bert | $90.51 \pm 0.16$ | $90.51 \pm 0.17$ | $-0.00$ | 73.2 | $90.49 \pm 0.17$ | $-0.02$ | 885.6 |
| | Roberta | $93.17 \pm 0.07$ | $93.14 \pm 0.05$ | $-0.03$ | 96.9 | $93.18 \pm 0.07$ | $+0.01$ | 886.8 |
| | Electra | $93.39 \pm 0.02$ | $93.32 \pm 0.09$ | $-0.07$ | 131.5 | $93.39 \pm 0.01$ | $+0.00$ | 933.5 |
| Amazon reviews | MPNet | $83.73 \pm 0.45$ | $83.52 \pm 0.32$ | $-0.20$ | 97.3 | $83.63 \pm 0.46$ | $-0.10$ | 365.2 |
| | Modern-Bert | $79.85 \pm 0.60$ | $78.52 \pm 1.04$ | $-1.33$ | 90.8 | $79.85 \pm 0.65$ | $-0.00$ | 340.0 |
| | Bert | $81.91 \pm 0.56$ | $81.45 \pm 0.91$ | $-0.46$ | 64.8 | $81.39 \pm 0.84$ | $-0.52$ | 307.4 |
| | Roberta | $84.33 \pm 0.36$ | $84.28 \pm 0.19$ | $-0.05$ | 89.0 | $84.33 \pm 0.21$ | $+0.01$ | 402.3 |
| | Electra | $85.10 \pm 0.60$ | $85.20 \pm 0.44$ | $+0.10$ | 60.3 | $85.06 \pm 0.64$ | $-0.04$ | 254.1 |
| AG News | MPNet | $93.22 \pm 0.10$ | $93.22 \pm 0.10$ | $+0.00$ | 139.2 | $93.20 \pm 0.09$ | $-0.02$ | 185.4 |
| | Modern-Bert | $91.61 \pm 0.06$ | $91.60 \pm 0.08$ | $-0.01$ | 150.2 | $91.63 \pm 0.07$ | $+0.01$ | 194.4 |
| | Bert | $93.03 \pm 0.09$ | $93.03 \pm 0.13$ | $+0.00$ | 128.4 | $93.04 \pm 0.12$ | $+0.01$ | 199.3 |
| | Roberta | $93.35 \pm 0.07$ | $93.33 \pm 0.03$ | $-0.02$ | 142.1 | $93.33 \pm 0.05$ | $-0.02$ | 193.2 |
| | Electra | $92.27 \pm 0.10$ | $92.27 \pm 0.13$ | $+0.00$ | 143.0 | $92.26 \pm 0.11$ | $-0.01$ | 185.9 |
| **Mean** | | | | $-0.14$ | 106.0 | | $-0.04$ | 471.9 |

The results may be seen in Figure 3 and Table 1. Overall, we found that it was possible to optimize towards prototypes which had near perfect similarity scores to the learned prototypes as their projected counterparts, resulting in a relatively insignificant accuracy performance difference (Opt.=0.03% drop v. Proj.=0.01%). Notably, AG News saw similarity drop-off most at the beginning of optimization (0.84 cosine similarity v. 0.99 nearest neighbour projection), signally a harder optimization problem, but nevertheless it progressed to the same similarity as the projection variant over 20 iterations and matched its accuracy performance, but with prototypes which were on average 25% shorter. Across all three datasets the change in prototype intelligibility was highly significant, as they dropped from an average of 682.3 characters in length using nearest neighbor projection, to just 95.5 with the optimized variant, meaning our prototypes are on average just 13.9% the size compared to the baseline. For more detailed results see Appendix C.

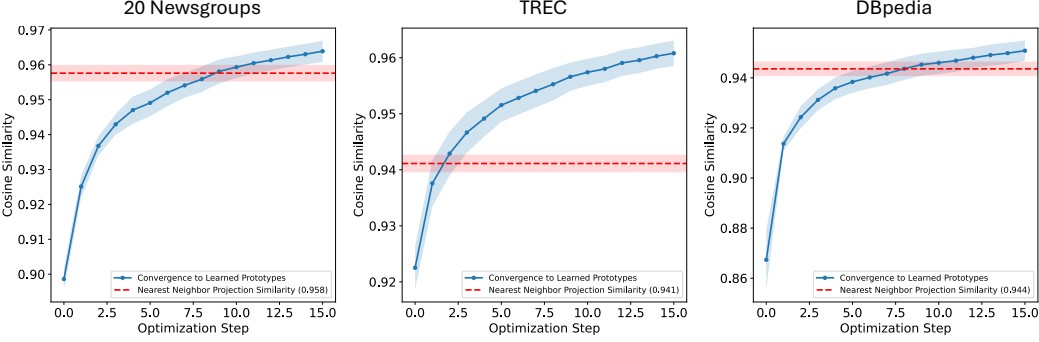

Figure 4: Optimization curves on the accuracy experiment: We found across all three datasets that it was possible to discover prototypes with higher similarity to the learned prototypes after just 15 iterations on average. Initially, there is a significant drop in similarity, but the optimization curves quickly achieve better similarity than the nearest neighbour prototypes which are typically used in the literature. Overall, this resulted in a significantly smaller accuracy drop for the optimized prototype networks compared to the projection-based ones (Opt=0.78% drop v. Proj=1.12% drop). All results are averaged across models, and show mean and standard error across 6 random seeds.

Table 2: Accuracy experiment results: The learned prototype network accuracy is displayed before any projection or optimization, alongside the subsequent accuracy, delta drop in accuracy, and cosine similarity to the learned prototypes using both our optimization approach, and prototype projection. Overall, there is significantly higher cosine similarity to the learned prototypes after using our optimization approach, and a smaller $\Delta$ drop in accuracy.

| Dataset | Model | Learned Acc. | Optimized Acc. | $\Delta$ | $s(\cdot,\cdot)$ | Projected Acc. | $\Delta$ | $s(\cdot,\cdot)$ |
|---|---|---|---|---|---|---|---|---|
| 20newsgroups | MPNet | $83.60 \pm 0.15$ | $83.40 \pm 0.07$ | $-0.20$ | $0.97$ | $83.03 \pm 0.19$ | $-0.58$ | $0.95$ |
| | Modern-Bert | $70.51 \pm 1.72$ | $69.65 \pm 1.83$ | $-0.87$ | $0.98$ | $68.89 \pm 1.79$ | $-1.62$ | $0.98$ |
| | Bert | $81.27 \pm 0.40$ | $81.05 \pm 0.35$ | $-0.21$ | $0.97$ | $80.51 \pm 0.43$ | $-0.76$ | $0.95$ |
| | Roberta | $80.42 \pm 1.50$ | $80.14 \pm 1.52$ | $-0.28$ | $0.98$ | $80.07 \pm 1.50$ | $-0.35$ | $0.98$ |
| | Electra | $69.18 \pm 1.76$ | $67.95 \pm 2.50$ | $-1.22$ | $0.97$ | $67.77 \pm 2.56$ | $-1.40$ | $0.97$ |
| Dbpedia | MPNet | $87.77 \pm 0.56$ | $87.83 \pm 0.47$ | $+0.06$ | $0.99$ | $87.57 \pm 0.32$ | $-0.20$ | $0.98$ |
| | Modern-Bert | $84.11 \pm 0.31$ | $84.10 \pm 0.33$ | $-0.01$ | $0.98$ | $83.75 \pm 0.34$ | $-0.36$ | $0.97$ |
| | Bert | $88.22 \pm 0.25$ | $88.48 \pm 0.26$ | $+0.26$ | $0.96$ | $88.37 \pm 0.45$ | $+0.15$ | $0.94$ |
| | Roberta | $87.63 \pm 0.46$ | $87.26 \pm 0.72$ | $-0.37$ | $0.98$ | $86.07 \pm 1.09$ | $-1.56$ | $0.96$ |
| | Electra | $86.34 \pm 0.26$ | $86.36 \pm 0.24$ | $+0.02$ | $0.98$ | $86.33 \pm 0.23$ | $-0.01$ | $0.98$ |
| Trec | MPNet | $91.60 \pm 0.76$ | $91.40 \pm 0.76$ | $-0.20$ | $1.00$ | $91.33 \pm 0.81$ | $-0.27$ | $0.99$ |
| | Modern-Bert | $73.53 \pm 0.07$ | $72.53 \pm 0.35$ | $-1.00$ | $0.99$ | $72.73 \pm 0.64$ | $-0.80$ | $0.99$ |
| | Bert | $91.07 \pm 0.55$ | $90.80 \pm 0.23$ | $-0.27$ | $0.98$ | $90.87 \pm 0.29$ | $-0.20$ | $0.97$ |
| | Roberta | $86.93 \pm 2.29$ | $86.93 \pm 1.91$ | $+0.00$ | $0.99$ | $86.27 \pm 2.32$ | $-0.67$ | $0.98$ |
| | Electra | $86.07 \pm 1.16$ | $85.93 \pm 1.09$ | $-0.13$ | $0.99$ | $85.73 \pm 1.35$ | $-0.33$ | $0.98$ |
| Mean | | | | $-0.29$ | $0.98$ | | $-0.60$ | $0.97$ |

## 4.2 CLOSING THE ACCURACY GAP DUE TO PROTOTYPE PROJECTION

In this section we test if our method can improve the classification performance when prototype projection results in a significant degradation. We again utilize the same models, but this time consider the datasets TREC (Li & Roth, 2002), DBpedia (Zhang et al., 2015), and 20 Newsgroups (Lang, 1995). We selected these datasets due to their relative difficulty which is necessary to maximize the effect of accuracy loss due to prototype projection (Chen et al., 2019). We set the number of prototypes in each model equal to the number of classes, and under-sampled DBpedia to its 20 most difficult to classify classes, to maximize the number of seeds reported. We optimized for 15 iterations per prototype. Additional details may be found in Appendix A and B. All tests were run across 6 seeds, with the means and standard errors reported.

The results may be seen in Figure 4 and Table 2, where the commonly reported artifact of accuracy loss due to prototype projection was observed. We found that on average it was possible to optimize towards prototypes which were more similar to the learned prototype vectors than the nearest neighbour projections, this resulted in both higher cosine similarity scores (Opt=0.96 v. Proj=0.95), and accuracy metrics (Opt.=0.78% drop v. Proj.=1.12% drop). Notably, TREC benefited the most from the optimization with the highest increase in cosine similarity and accuracy, showing that the prior is a reasonable proxy for the latter.

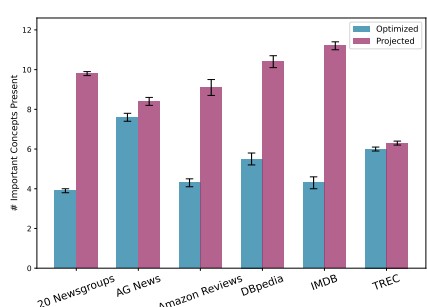

## 4.3 QUALITATIVE ANALYSIS

Recent studies have shown LLMs can function as a reliable proxy for user evaluation (Cui et al., 2024), hence, we used `claude-3-7-sonnet-20250219-v1:0` to analyze our systems in an "LLM as a judge" setup (Gu et al., 2024). We sampled 100 test instances from each dataset and model pairing, across seeds (i.e., 10,800). Then, we ran inference and took the closest prototypes in both model types (LLM-optimized and training data projection). For

Figure 5: Qualitative test: Our method preserved 57% relevant concepts to the test instances, despite being only 11% of the length of the projected prototypes. This highlights that our process has a positive tradeoff between the prototype length and concept preservation.

each, a prompt was constructed which included the test instance and both prototypes. We instructed

the LLM to list all high-level concepts in both prototypes which could be used for classification, and then decide which was most similar. The results are in Figure 5, where our method was flagged as having 57% of the concepts on average compared to projection (5.3±0.2 v. 9.2±0.2), despite being only 11% the length. This showed that our method generates prototypes which are nearly 10x shorter while preserving most of the important concepts. We observed no significant difference in the decisions of which prototype was most similar to the test example (Opt.=49% v. Proj.=51%), illustrating the important concepts for similarity were preserved. Put another way, the vast majority of the seemingly important concepts identified by the LLM were actually not relevant for assessing similarity to the test instances. Hence, although our approach did lose almost half the concepts, they were spurious or irrelevant features which would only serve to confuse human users and increase cognitive load (Doshi-Velez & Kim, 2017). For the full prompt see Appendix G.

## 5 DISCUSSION

We proposed a paradigm shift in how to train prototype neural networks, which works by visualizing the learned prototypes through an optimization process involving foundation models rather than simply projecting them to nearest neighbours in the training data. Our results showed that we can drastically simplify prototypes down to their core concepts without sacrificing performance, and in the cases where projection does result in performance drops, we can help mitigate this to improve the networks. The reduction in prototype length will drastically reduce cognitive load for users (Matthews & Folivi, 2023), and increase the accessibility of the system (Darling-White & Polkowitz, 2023).

During early testing we initially used Claude-3.7-Sonnet as our optimizer LLM, and it produced even better results as well as being 10x faster, especially when queried in parallel (up to 50 times). However, we opted to use `Meta-Llama-3-8B-Instruct` because of two reasons. First, the model is far more accessible for researchers, as long as the lab has one GPU, it will work with no API cost. Secondly, we found other open source models such as `Qwen` 7B-70B in size gave no great benefit, but just slowed down generation due to being more computationally expensive.

A possible criticism of our work is that we only focused on text domains, in future work it would be interesting to replicate our results in an image domain with part-prototype networks using stable diffusion (or other suitable models) in place of our LLMs.

## REPRODUCIBILITY STATEMENT

The code to reproduce the results was given with the submission. Follow the `readme.txt` file and the results will reproduce. An API subscription to anthropic is required to reproduce the results of our qualitative test, and access to multiple GPUs is recommended.

## ETHICS STATEMENT

One of the potential ethical considerations of using prototypes for explanations is that the examples used must persevere privacy and not be offensive or unethical in any sense. However, provided the LLM used for optimization is appropriately aligned, the probably of producing offensive content is highly mitigated. Moreover, as the number of prototypes in our models is quite small, it is more easily screened before deployment, which helps significantly compared to more traditional CBR approaches that may use the whole dataset or a large portion.

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

## A    TRAINING HYPERPARAMETERS

The models used were BERT, RoBERTa, and ELECTRA backbones with prototype-based classification heads, the final block on each was finetuned. Datasets were six text classification datasets: IMDB, Amazon Reviews, AG News, TREC, DBpedia, and 20 Newsgroups.

**Training Epochs:** Dataset-specific epochs: IMDB and Amazon Reviews (5 epochs), AG News (10 epochs), DBpedia (15 epochs), 20 Newsgroups (20 epochs), and TREC (100 epochs).

**Prototypes per Class:**

- 3 prototypes: IMDB, Amazon Reviews, AG News
- 1 prototype: TREC, DBpedia, 20 Newsgroups

**Architecture:**

- Input sequence length: 256 tokens
- Prototype dimension: 256
- Backbone models fine-tuned with prototype layers

**Optimization:**

- AdamW optimizer (lr=3e-4, weight decay=0.01)
- Batch sizes: 32 (training), 128 (validation/test)
- Train/validation split: 95/5 stratified

**Loss Components:** Classification loss with prototype regularization terms. Loss weights: $\lambda_{p1} = 0.1$, $\lambda_{p2} = 0.01$, $\lambda_{p3} = 0.01$.

**Experimental Setup:** 6 random seeds (0-5), multi-GPU training across 4 CUDA devices, and early stopping based on validation accuracy.

## B    DATASET PREPROCESSING

This appendix details the preprocessing steps applied to each dataset used in our experiments. Some datasets were under-sampled to make training tractable, or filtered down to their most difficult classes to enhance the phenomenon of prototype projection causing accuracy drops.

### B.1    DBPEDIA

- Combined training, validation, and test splits from the original dataset
- Selected 20 target classes from the full set of categories [185, 166, 159, 57, 160, 168, 146, 198, 123, 38, 1, 73, 36, 56, 54, 215, 39, 128, 90, 171]. These were selected by training a standard BERT classifier on the whole dataset and picking the 20 classes with the worst classification performance.
- Applied a 90:10 train-test split to the filtered data

### B.2    AMAZON REVIEWS (CELL PHONES & ACCESSORIES)

- Converted 5-point ratings to 3-class sentiment labels: ratings 4-5 mapped to Positive, rating 3 to Neutral, and ratings 1-2 to Negative
- Filtered out invalid text entries (null, empty, or whitespace-only reviews)
- Randomly sampled 100,000 examples from the filtered dataset
- Used a 90:10 train-test split

### B.3    OTHER

The 20 Newsgroups, TREC, IMDB, and AG News required no pre-processing.

## C  FULL OPTIMIZATION RESULTS

In this section we include figures of the full optimization results from Section 4 in Figures 6 and 7.

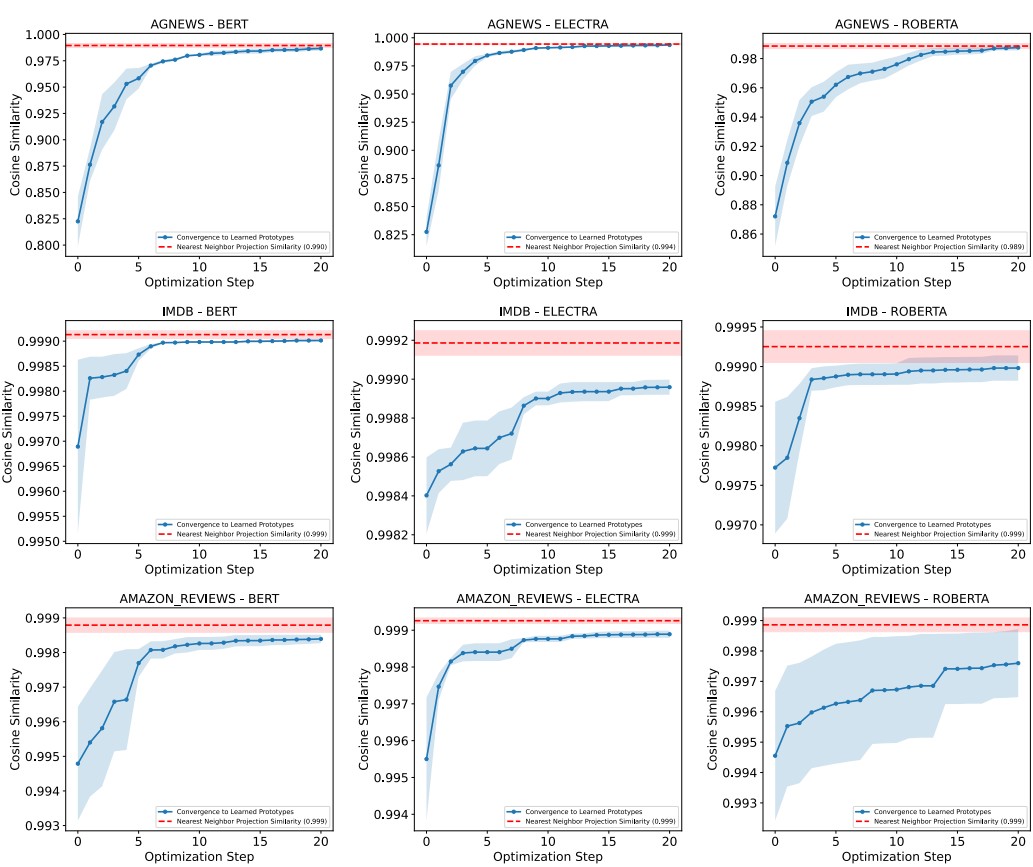

Figure 6: Full results for interpretability experiment.

## D  OPTIMIZATION EXAMPLE

Table 3 shows an example of what the textual optimizations look like in 20 Newgroups using BERT. This examples started off particularly low in similarity, but quickly converged to a much higher value.

## E  COMPUTATIONAL COST

Preliminary evaluations compared the efficacy of proprietary black-box API services against locally executed models on GPU infrastructure. The latter approach was selected to prioritize reproducibility and mitigate operational costs for the research community. Regarding latency benchmarks, API-based inference demonstrated the capacity to execute 20 optimization iterations in approximately 60 seconds (excluding Chain-of-Thought reasoning). In contrast, our local experimental configuration—utilizing a standard CUDA-enabled device with 24GB of VRAM—exhibited a latency increase of approximately one order of magnitude ($10\times$)4. However, it is critical to contextualize this computational overhead: the optimization of prototypes represents a non-recurring initialization cost. Once optimized, the model parameters are fixed, ensuring that this initial computational expense is amortized over the model's life-cycle and does not impact subsequent inference efficiency.

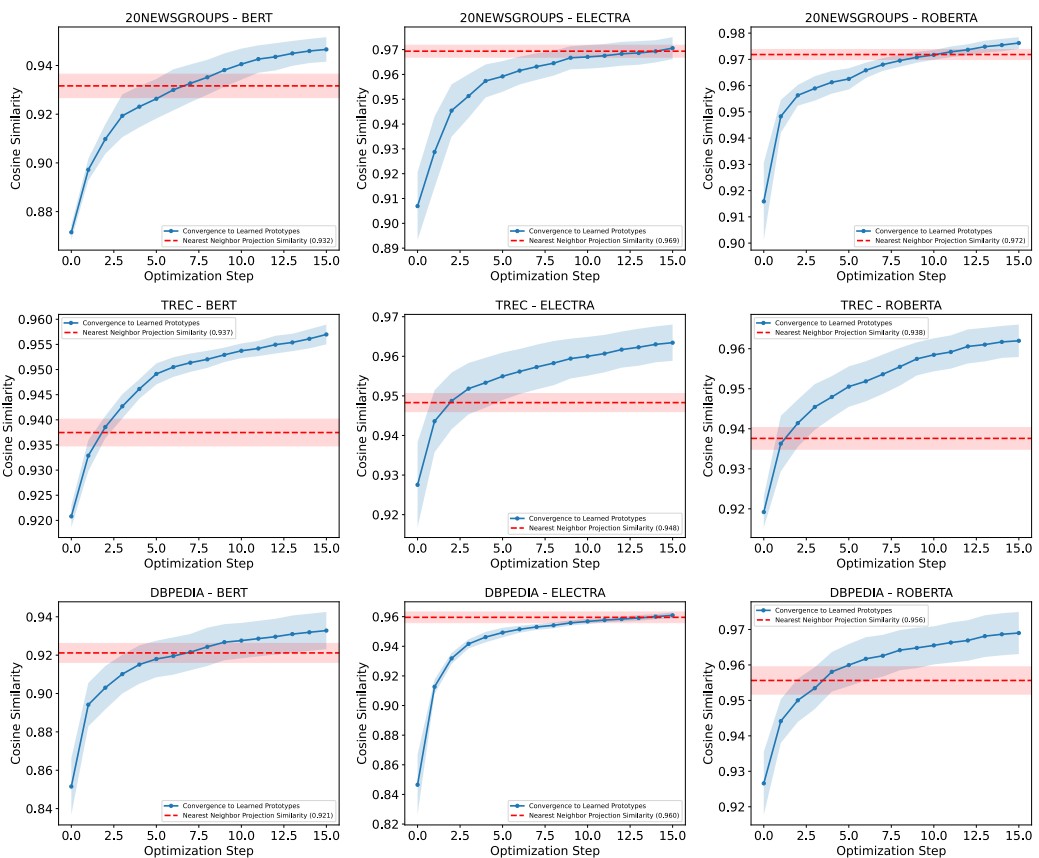

Figure 7: Full results for accuracy experiment.

## F  LLM USAGE

LLMs were mainly used to help edit experimental code in a collaborative setup and assist with polishing the manuscript before submission. Their usage in writing was extremely minimal. They had no role in research ideation. They had no role in retrieval or discovery of related work.

## G  PROMPT TEMPLATES

### G.1  PROMPT FOR OPTIMIZATION INITIATION

I am trying to identify a prototypical example from the `{dataset_name}` dataset.

The prototype should represent a typical example of a `{description}`'. The following examples are very similar to the real prototype: `{examples_str}`

Based *only* on these examples, please generate a Python list containing exactly `{num_guesses_to_generate}` distinct, concise, and relevant phrases or sentences that you believe also capture the core concepts in these examples in a prototypical sentence.

Each phrase should be a potential textual description of the prototype and its core concepts.

Your output must be ONLY a single Python list of strings. For example: ["first candidate phrase", "second candidate phrase", ..., "tenth candidate phrase"]

Generated Python list:

Table 3: Example of prototype optimization best guesses on 20 Newsgroups using BERT.

| Iteration | Current Best Guess | Cos Sim. |
|:---:|:---|:---:|
| 0 | There are concerns about the rate at which two substances interact. | 0.59 |
| 1 | A sports highlight from a recent championship game | 0.67 |
| 2 | A sports news article about a recent match | 0.70 |
| 3 | A sports highlight from a recent tournament | 0.72 |
| 4 | A sports highlight from a recent tournament | 0.72 |
| 5 | A sports highlight from a recent tournament | 0.72 |
| 6 | Article about a recent sports championship win | 0.73 |
| 7 | Article about a recent sports championship win | 0.73 |
| 8 | Championship news headlines being released. | 0.75 |
| 9 | Championship news headlines being released. | 0.75 |
| 10 | Up-to-date sports news about championships. | 0.76 |
| 11 | Championship news from last week. | 0.82 |
| 12 | Championship news from last week. | 0.82 |
| 13 | Championship news being released recently now. | 0.86 |
| 14 | Recent news from championship headlines. | 0.87 |
| 15 | Recent news from championship headlines. | 0.87 |

### G.2 META PROMPT FOR OPTIMIZATION

You are a helpful assistant to a data scientist.

We are working together to try find a text sequence which perfectly maps to a learned black box prototype vector in the latent space of a language model. In doing so, we are querying you repeatedly in an optimization loop. This is one of those loops.

I will show you the current {num_neighbors} text sequences you generated perviously, and their cosine similarity to the prototype. The closer the similarity is to 1, the better the guess is, because it's more similar to the prototype, the similarity ranges from -1 to 1. Our goal is to find a text sequence which perfectly maps to the prototype and gives a score of 1.

Here are the current {num_neighbors} text sequences you have generated previously in a query: {population} Their similarity scores are: {[round(c, 2) for c in np.array(distances).flatten()]}

Can you suggest another {num_neighbors} guesses which are closer to 1?

The prototype should represent a short, prototypical example of a positive '{description}'.

If a lot of your guesses are similar, you should try diversify them to avoid getting stuck in a local minimum, you can try vary the length, or even take random guesses. Here are some close training data neighbors of the black-box prototype to help you get some variety in your guesses: {training_examples}

Respond ONLY with your guesses as a Python list of strings.

For example:

["first guess", "second guess", "...", "last guess"]

It is extremely important you follow this format exactly.

### G.3 PROMPT FOR QUALITATIVE ANALYSIS

You are analyzing prototypes used by a neural network classifier that uses cosine similarity for classification on the {dataset} dataset. The prototypes are being used to classify the test instance based on their cosine similarity to it, your job is to help us analyze the prototypes.

Test Instance to Classify: {test_text}

Stage A Prototype: {stage_a_proto}

Stage B Prototype: {stage_b_proto}

Please analyze these prototypes and the test instance:

1. First, identify ALL high-level concepts in the Stage A prototype that could be used by a classifier.

2. Do the same analysis for the Stage B prototype.

3. Based on cosine similarity principles, determine which prototype would be most similar to the test instance.

You should be comprehensive, you don't need to have the same number of concepts for both prototypes, it is ok for one to have many more concepts.

Provide detailed reasoning for your analysis, then output a JSON object with the following structure: "'json {{

"stage_a_concepts_count": <integer>,

"stage_b_concepts_count": <integer>,

"most_similar_prototype": "<'stage_a' or 'stage_b'>",

}} "'

The length and detail level of the prototypes do NOT matter for classification purposes, do not consider them in your analysis, only focus on high-level concepts for the classification.

