# OpenReview forum: "Finding Better Prototypes For Interpretable Text Classifiers With LLM Optimization"
_ICLR.cc/2026/Conference — Submitted to ICLR 2026_

### Official Review · Reviewer_3GPv · 2025-10-17

**Soundness:** 1
**Presentation:** 1
**Contribution:** 1
**Rating:** 2
**Confidence:** 2

**Summary:**

This paper presents a method to improve the interpretability of prototype neural networks by improving prototypes via large language models (LLM). Existing methods project learned prototypes onto nearest training examples, leading to performance drops and overly specific representations. This paper addresses this issue by using LLMs to optimize and discover better textual prototypes. To accomplish this, latent prototype vectors (numerical representation) are first learned on a text classification task. Textual representations for each of these latent prototype vectors are then found from the training text corpus based on cosine similarity. These texts are improved via LLM to further minimize the cosine similarity in the same latent space. With these optimized prototypes, text classification accuracy is on par compared to simply projected prototypes. However, from a qualitative evaluation, prototype quality appears to be better in LLM optimized prototypes.

**Strengths:**

The idea of using LLM to improve prototypes is interesting, and I find it novel.

**Weaknesses:**

First and foremost, I'm not convinced about the idea of using LLMs (plural) to improve prototypes. LLMs are black boxes with even more parameters than the algorithm that the paper is trying to interpret/visualize. I think the fundamental assumption of this paper is that LLMs are sufficiently intelligent that one can trust what LLM suggests, which I don't necessarily agree with. Moreover, I don't understand if it is necessary to employ multiple, repeated LLM inferences for training a simple text classifier.

Also, if I understood correctly, the latent prototype vectors will stay the same during the LLM optimization, which means that the LLMs will not contribute to the improvement of the text classification accuracy. If so, LLMs' role appears to be to simply tweak and wordsmith the textual representation of those prototypes. I don't know if this would necessarily lead to an "improvement" of interpretability.

Additionally, this paper will benefit a lot from improving the presentation. The current presentation does not elaborate on the core idea and concepts effectively, which required me multiple readings to understand what exactly was going on. Mathematical equations are also not very helpful due to insufficient rigor and details.

To be fair, I might be misunderstanding something about the scientific contribution of this work. However, in its current form, the rationale behind the use of LLMs to improve prototypes is unclear and how the optimization was implemented and conducted is also unclear. Hence, the low rating.

**Questions:**

- Equation 7: Where does the LLM \mathcal{L} play a role in this objective function? I suppose t is the output of LLM, but I'm not sure.
- Section 4.3: I don't understand the rationale here. Especially, Figure 5: # of important concepts present in a prototype is much lower for optimized prototypes--> isn't that a bad thing?

---

> ### Author Response · Authors · 2025-11-20
> **Author(s) Rebuttal**
>
> Many thanks for your review, we now respond to all you points.
>
> **Not convinced about the idea of using LLMs to improve prototypes…LLM inferences for training a simple text classifier.:** Thanks for the questions, but we feel there may me a slight misunderstanding here. Firstly, to curb confusion we are not training the classifier with the LLM, we are simply using the LLM as a tool to take guesses as to what text best maps to a classifier’s learned prototype(s) in latent space AFTER training, and then projecting the best guess into the model. We discussed this in Section 3.2.3. Depending on the prototype quality, this does affect model accuracy (note the same is true for baselines), but we are not retraining the weights of the encoder at this point.
> Second, **we are not blindly trusting the LLM**. We use an objective metric (cosine similarity) to measure how good the LLM’s guesses are, the LLM is just a tool to generate text.
> Third, you don’t need multiple LLMs in parallel, but it increases the diversity of guesses each iteration, which helps convergence as Yang et al. [1] discuss. We included a new discussion in Section 3.3 about all this.
>
>
> **Also, if I understood correctly… I don't know if this would necessarily lead to an "improvement" of interpretability:**   We think there may be another misunderstanding here, allow us to explain. In prototype learning you typically learn black-box prototype vectors, which are projected onto training instances to map them to real inputs, this typically makes the accuracy drop [2]. Standard prototype methods also do not “contribute” to improvement in text classification accuracy, they typically hurt it [2], which is one problem we studied (and partly solved) here. **Our variant finds better alternatives to the training data which make accuracy drop less**.
> Regarding interpretability, the prototypes are 11% the size of baselines, a reduction that directly lowers the intrinsic cognitive load for users [3]. By minimizing the span of text, we reduce the demand on working memory, which is widely cited as useful by the interpretability community [4].
>
>
> **Presentation:**  Thanks for this feedback, we iterated our Methods section using your suggestions to improve presentation. If there is anything specific the reviewer can point to, we can further iterate on that section, we would greatly appreciate your help. We added Section 3.3 and Appendix E to help rationalize our hyperparameter choices more.
>
>
> **The rationale behind the use of LLMs to improve prototypes.. and how the optimization was implemented… is also unclear:** LLMs are the best current technology to optimize textual data [1], since prototypes in our domain are text, it makes sense to use LLMs to find the best prototypes possible. We regret the reviewer found the writing unclear, we have taken your suggestions on board and tried to rewrite the Methods section, we would truly appreciate specific suggestions to help the presentation, just let us know which part was confusing.
>
>
> **Equation 7:**  It represents a guess for the prototype, which is generated by an LLM, we have reworded the follow-up sentence to state that more explicitly, thanks for that.
>
>
> **Section 4.3:** No it’s not bad, we regret the lack of clarity in the writing here. Because task performance of the LLM-as-a-judge stayed the same despite this loss in concepts, this shows we effectively eliminated spurious/irrelevant concepts during optimization with the LLM(s) that would only serve to mislead users and increase cognitive load [3, 4]. We have made this clearer in Section 4.3., thanks for letting us know.
>
>
>
> Please let us know your updated thoughts after this response to your comments, thank you.
>
> ***
>
> [1] Yang, C., Wang, X., Lu, Y., Liu, H., Le, Q.V., Zhou, D. and Chen, X., 2023, September. Large language models as optimizers. In The Twelfth International Conference on Learning Representations.
>
> [2] Chen, C., Li, O., Tao, D., Barnett, A., Rudin, C. and Su, J.K., 2019. This looks like that: deep learning for interpretable image recognition. Advances in neural information processing systems, 32.
>
> [3] Miller, G. A. (1956). The magical number seven, plus or minus two: Some limits on our capacity for processing information.
>
> [4] Doshi-Velez, F. and Kim, B., 2017. Towards a rigorous science of interpretable machine learning.

---

### Official Review · Reviewer_Mibe · 2025-10-29

**Soundness:** 3
**Presentation:** 3
**Contribution:** 2
**Rating:** 4
**Confidence:** 3

**Summary:**

This paper proposes using LLMs to directly optimize/generate prototypes that better reflect the learned representations. The experiments report more intelligible prototypes with performance comparable to projection baselines. This suggests a path to visualizing learned prototypes without relying on actual training instances.

**Strengths:**

•	The work addresses one key limitation of standard projection-based methods.

•	The work is well-motivated and important to ML interpretability.

**Weaknesses:**

- High computational cost. This method uses LLMs as optimizers, which require multiple iterations and parallel LLM inferences per prototype and are thus computationally expensive. The paper does not provide a detailed analysis of the computational costs, particularly for cases with large numbers of classes and prototypes.

- Lack of human evaluation. The qualitative analysis of the optimized prototypes relies solely on an "LLM-as-a-judge" framework. Without a user study or human annotations, there is no direct evidence that they are more intelligible or helpful for humans trying to follow the model’s reasoning.

- Limited domain generalization. The method is validated only on text. While image extensions are suggested, the paper offers no details on how to adapt the paradigm to other domains, where producing abstract representations may be harder than generating text.
________________________________________
Minor typo:
	Line 147-148: {X}_(i=0)^N --> {X}_(i=1)^N
	Line 339-340: datum --> data

**Questions:**

- Lack of stagnation analysis. How does the "optimizer" handle local minima, where the LLM repeatedly generates candidates that show little or no improvement? Did the authors observe this in practice, and what techniques (if any) were used to escape such minima?

- Choice of LLM. Why choose Meta-Llama-3-8B-Instruct as the optimizer model? Have you tried to use other LLMs? For example, would a less powerful model suffice? Would a more advanced model (e.g., GPT-5) yield even better prototypes?

- Depth of concept preservation analysis. In the qualitative analysis (Section 4.3), the authors report that the optimized prototypes preserve 57% of the concepts found in the projected prototypes. Could the authors elaborate on why this is sufficient to support the claim of "preserving most of the important concepts"? Furthermore, has any deeper analysis been conducted into the nature of the concepts preserved versus those lost at 43%? For example, do the preserved concepts tend to be the most critical for the classification decision, while the omitted ones are more secondary or contextual?

- Limited baseline comparison. The experiments mainly compare the LLM-based optimization method with the standard practice of projecting prototypes onto nearest neighbors. Why were additional text-classification baselines not evaluated?

---

> ### Author Response · Authors · 2025-11-20
> **Author(s) Response**
>
> Thank you for your review, we now respond to all your concerns.
>
> **Computation cost:** Sorry for neglecting to include this, that was an oversight. Initially we conducted tests comparing black-box APIs and locally run models on GPUs. We opted for the latter as it typically reduces costs for researchers and allows better reproducibility (especially as APIs can update with “hidden” wrapper modules). The time for e.g. 20 iterations on API models would typically be 60 seconds without CoT. For us running a basic CUDA with 24GB memory, it takes almost 10x longer. **Whilst it is expensive to optimize towards these prototypes initially, it is only required once**, then the model is stored and used normally. We now included these computational costs in Appendix E, thanks for the comment.
>
> **User evaluation:**  Plenty of work has shown the usefulness of similar explanations already [6]. Our prototypes are 11% the size of baselines, a reduction that directly lowers the intrinsic cognitive load for users [2], something the XAI community agrees is useful [3]. We have emphasized the usefulness of the approach to reduce cognitive load in the final Discussion, and how prior work has shown LLMs are a reasonable proxy for human experiments in Section 4.3 [4].
>
> **Domain generalization:** True, but this is a concern which could be voiced about almost all prototype interpretability papers, even the most seminal [5]. This is a great direction, but it is one for future research.
>
> **Lack of stagnation analysis:** Yang et al. [1] did this work before us (and we converged with their findings early on) and found that local minima where an issue. To overcome this, we (1) got the LLMs to produce multiple guesses per iteration, (2) used higher temperature=1.0, and (3) included random training examples in the prompt. In addition, to help more we also included special instructions in the prompt. For the latter, we said *“If a lot of your guesses are similar, you should try diversify them to avoid getting stuck in a local minimum, you can try vary the length, or even take random guesses”* ”—See Appendix F.2. In practice, all of these helped a lot. We now discuss this in Section 3.3, but initially did not include it as it simply reproduces prior work [1].
>
> **Justify LLM choice:** We initially used the API for Claude-3.7-Sonnet, and it produces even better results (10x faster too). However, we opted to use Meta-Llama-3-8B-Instruct because: (1) it is far more accessible for researchers (as long as the lab has one GPU, it will work with no API cost), (2) better open-source models such as Qwen 7B-70B gave no great benefit, and (3) as previously mentioned it allows full reproducibility. We now mention this in the Discussion.
>
> **Clarify Section 4.3:** We regret the lack of clarity in the writing here, because task performance of the LLM-as-a-judge stayed the same despite this loss in concept information, this shows we effectively eliminated spurious/irrelevant concepts that would only serve to mislead users and increase cognitive load [2, 3]. Hence, our prototypes have less concepts yes, but they are the ones which are actually important, we eliminated the spurious/irrelevant ones. We have made this clearer in Section 4.3., thanks for letting us know.
>
> **Why were additional text-classification baselines not evaluated:** They were, they are the “Learned” column in Table 1 and 2, they constitute 30 comparisons across the two tables in our updated experiments (now including two additional language models MPNet and ModernBERT).
>
> Thank you for the review and comments, we are eager to hear your thoughts after our response, thank you.
>
> ***
>
> [1]  Yang, C., Wang, X., Lu, Y., Liu, H., Le, Q.V., Zhou, D. and Chen, X., 2023, September. Large language models as optimizers.
>
> [2] Miller, G. A. (1956). The magical number seven, plus or minus two: Some limits on our capacity for processing information.
>
> [3] Doshi-Velez, F. and Kim, B., 2017. Towards a rigorous science of interpretable machine learning.
>
> [4] Cui, Z., Li, N. and Zhou, H., 2024. Can ai replace human subjects? a large-scale replication of psychological experiments with llms.
>
> [5] Chen, C., Li, O., Tao, D., Barnett, A., Rudin, C. and Su, J.K., 2019. This looks like that: deep learning for interpretable image recognition.
>
> [6] Chen, V., Liao, Q.V., Wortman Vaughan, J. and Bansal, G., 2023. Understanding the role of human intuition on reliance in human-AI decision-making with explanations.

---

### Official Review · Reviewer_6YW5 · 2025-11-01

**Soundness:** 1
**Presentation:** 2
**Contribution:** 1
**Rating:** 2
**Confidence:** 4

**Summary:**

The paper focuses on a specific form of interpretability: prototype based interpretability where the model explains its decisions by pointing to prototypes from the training data. The starting point of the paper is how the prototypes based explainability is typically done: learning the prototype vectors in the penultimate layer and then projecting them to the nearest training data point to generate the explanation. The paper points out that per the prototype theory of Rosch, prototypes should be abstract representations of the class, which is difficult to do for LLMs with large sequence lengths. The key idea of the paper is to use LLMs as optimization tools which can help build more concise, simple and general prototypes. The methodology is quite similar to the traditional work on prototype based learning where the training objective is a combination of different losses that ensure good classification accuracy and learning of distinct, grounded prototypes (Section 3.1). The paper then refines the prototype by using a LLM that summarizes the nearest neighbors of the initial prototype.

**Strengths:**

1. Model interpretability is indeed an important topic and lack of understandability is a large blocker is building trust in LLMs.
2. The core idea of the paper is grounded in the prototype theory. It does indeed make sense that prototypes should not focus on spurious and overly specific features but represent more abstract concepts contained within the class.

**Weaknesses:**

1. The writing of the paper can be improved to add key details. (i) What is the dataset level description and why is it needed? (ii) Instead of using multiple “meta-prompts” that consist of a random sample of nearest neighbors, why not use a single meta-prompt that uses all the neighbors? (iii) How is the number of nearest neighbors and the number of meta-prompts determined? Given a new dataset, should these be treated like a hyperparameter? If yes, what should the optimization objective be?  (iv) In line 292, what is the difference between a single LLM operating on all input data vs different LLMs operating on different sets of the training data?
2. It is not clear how the idea would generalize to domains where its not the words like “spoof” and “technical quality” that are important, rather, its is the operators surrounding the words that have more importance, e.g., negation words like “not”. Is it possible that the prototypes will end up ignoring these small yet highly influential words? Some discussion that connects the makeup of prototypes to linguistic features would add a lot more weight to the paper’s contributions.
3. The proposed solution seems to be restricted to simple classification based tasks (AG News, IMDB Movie Reviews, Amazon Reviews, 20 Newsgroups) and is tested on relatively simpler models like BERT and RoBERTa. The paper should discuss if, and how, the method would be extended to generation based tasks like summarization and if we expect it to work on instruction tuned LLMs like LLaMA and Qwen.
4. It’s not clear what the prototypes add in terms of explainability for the end-user. Agreed that the prototypes learnt here are shorter, but what does that add for the end-user? The paper should provide some evaluation with humans showing that the prototypes learned here are actually helpful, e.g., perhaps they help users identify wrong classifications or remove poisonous data points.

**Questions:**

1. Please see the questions in W1
2. Why are the datasets different between Fig 3 / Table 1 and Fig 4 / Table 2?

---

> ### Author Response · Authors · 2025-11-20
> **Author(s) Rebuttal**
>
> Thank you for taking the time to write a thoughtful and in depth review, we now respond to all your comments:
>
> **Explain dataset level description:**  It is given in Appendix F.1, it is just one line telling the LLM what the dataset is, we empirically found it improved the initial guesses.
>
> **Why not use a single meta-prompt that uses all the neighbors:** Yang et al. [1] found more than 2-3 didn’t help (and we converged with these findings), so we limited our queries to the same.
>
> **NNs, meta-prompts, and optimization objective:**    These are hyperparameters, if you use too many NNs then the examples you show the LLM can be so far from the learned prototype that it disrupts optimization, if you use too little then there is not enough diversity to avoid early local minima, the numbers were empirically determined via a grid-search and will always be dataset specific (generally 20 in our tests).  The optimization objective Eq. (7) never changes.  We added this discussion to Section 3.3.
>
> **Single LLM on all input data vs LLMs on different sets:**    Both end in a similar result, but the latter is quicker because you get more guesses each optimization iteration [1]. We chose to do the latter because our LLM could only produce 10 guesses per iteration before running into errors in formatting the output list. We found these results early on, but omitted them from the paper as they simply duplicate prior findings [1]. We note this in our new Section 3.3.
>
> **Negation words in generated text:** We regret there seems to be a possible misunderstanding; this is not a concern as the optimization will only keep new solutions if they are more similar to the learned prototype. If these new guesses contain operators like “not”, then we will keep them, otherwise they will be dropped. We observed to bias in the LLM towards using these words or not. We did a qualitative analysis similar to what you ask for already in Section 4.3, which showed we maintained all the important linguistic features **for judging similarity**. Plenty of our prototypes contained words like “not” or similar negation words, we are happy to open source all of this if published so people can view all the prototypes.
>
> **GenAI extensions:** Thanks for the suggestion. We restricted ourselves to classification because this is what all prior work has done, and we are building on that, summarization is (we strongly argue) a very different research topic which would require very different solutions. For example, only one paper we are aware of has delt with prototypes and LLMs in a generative task [5], and it is quite different to typical classification solutions. Early on we did use Qwen and Llama to do the same classification tasks, but their performance was not competitive. We briefly note this now in Footnote 1. **We also added the more modern text classifiers MPNet and Modern-Bert** (released Dec. 2024) to our results.
>
> **Demonstrate utility of your prototype method:** Plenty of excellent work has shown the usefulness of similar explanations already to humans [6]. Our prototypes are 11% the size of these baselines, a reduction that directly lowers the intrinsic cognitive load for users [2], something the XAI community agrees is useful [3]. We have now emphasized the usefulness of the approach to reduce cognitive load in the final Discussion, and how prior work has shown LLMs are a reasonable proxy for human experiments in Section 4.3. [4].
>
> **Figures:** We used datasets with long text for Section 4.1 to demonstrate we could reduce the length of prototypes. We used datasets which were more complex for Section 4.2 because they naturally demonstrated a larger drop in accuracy after projection, which in turn allowed us to demonstrate we could make up for this with our method. The datasets served the purpose of each test (interpretability and accuracy, respectively), they also allowed testing on 6 datasets rather than 3, which shows better generalizability.
>
> We hope these responses make the reviewer reconsider their view of our work, and we eagerly await their response, thank you.
>
> ***
>
> [1] Yang, C., Wang, X., Lu, Y., Liu, H., Le, Q.V., Zhou, D. and Chen, X., 2023, September. Large language models as optimizers. ICLR
>
> [2] Miller, G. A. (1956). The magical number seven, plus or minus two: Some limits on our capacity for processing information. Psychological Review.
>
> [3] Doshi-Velez, F. and Kim, B., 2017. Towards a rigorous science of interpretable machine learning. arXiv preprint arXiv:1702.08608.
>
> [4] Cui, Z., Li, N. and Zhou, H., 2024. Can ai replace human subjects? a large-scale replication of psychological experiments with llms.
>
> [5] Bodla, K.V. and Yang, H., 2025. Protocode: Prototype-Driven Interpretability for Code Generation in LLMs.
>
> [6] Chen, V., Liao, Q.V., Wortman Vaughan, J. and Bansal, G., 2023. Understanding the role of human intuition on reliance in human-AI decision-making with explanations.

---

### Author Response · Authors · 2025-11-20
**Dear AC and Reviewers**

We thank the reviewers for their thoughtful and detailed feedback. We were encouraged to see that Reviewer 6YW5 appreciated the work’s strong grounding in prototype theory, Reviewer Mibe found the motivation to address the limitations of standard projection methods important, and Reviewer 3GPv found the core paradigm of using LLMs for prototype optimization to be novel.

We recognize that our initial manuscript lacked clarity in several key areas, particularly regarding the qualitative analysis, the justification for human utility, and why Generative AI was not included as a baseline. We have uploaded a revised manuscript (changes in red) that addresses these concerns. We summarize the primary changes below:

* **Qualitative Analysis & Concept Reduction:** Reviewers Mibe and 3GPv raised critical questions regarding Section 4.3—specifically, whether the loss of 43% of "concepts" in our optimized prototypes was a negative outcome. We realize our original explanation was insufficient. We have revised Section 4.3 to clarify that this reduction represents the successful filtering of noise rather than the loss of signal; we know this because predictive performance in the LLM-as-a-judge task was the same for both the baseline and our more concise optimized variants. Put another way, the test shows our method removes a lot of concepts that “seem important”, but actually are not, and only serve to increase user cognitive load and mislead [1, 2].
* **Human Utility:** Reviewers 6YW5 and Mibe asked about the usefulness of our method for human users. Many excellent studies have already shown the utility of similar explanations for appropriate reliance and model improvement [4], we have expanded our discussion in Section 5 to explain how the 86% reduction in prototype size directly serves to lower cognitive load [1, 2], which the ML community agrees is useful [3]. Additional user studies are not required to verify this, but we agree it is a nice direction for future work.
* **Expanded Baselines:** In response to Reviewer 6YW5, we have significantly expanded our baseline comparisons. Our early unreported experiments (now noted in Footnote 1) showed that LLMs were not competitive here, so we have added two additional state-of-the-art classifier language models to our evaluations—including ModernBERT (released Dec 2024)—to bolster our results. We are not focused on Generate AI here as that requires very different solutions beyond scope [5]. Note some results have changed, but all trend the same as before.
* **Reproducibility & Details:** To address general requests for more training specifics, we have added Section 3.3 and Appendix E.

We believe these revisions directly address the main objections raised. We are confident that this work makes a substantial contribution to the prototype literature by introducing a paradigm that improves accuracy while reducing complexity, and we would be thrilled to present this work at ICLR!

***

[1] Miller, G. A. (1956). The magical number seven, plus or minus two: Some limits on our capacity for processing information. Psychological Review.

[2] Lombrozo, T. (2007). Simplicity and probability in causal explanation. Cognitive Psychology.

[3] Doshi-Velez, F. and Kim, B., 2017. Towards a rigorous science of interpretable machine learning. arXiv preprint arXiv:1702.08608.

[4] Chen, V., Liao, Q.V., Wortman Vaughan, J. and Bansal, G., 2023. Understanding the role of human intuition on reliance in human-AI decision-making with explanations.

[5] Bodla, K.V. and Yang, H., 2025. Protocode: Prototype-Driven Interpretability for Code Generation in LLMs.

---

### Meta-Review · Area_Chair_JS7Z · 2026-01-08

**Summary:**

This work proposes to use LLMs for prototype optimisation in the context of prototype neural networks training. Experiments suggest the prototypes produced are reduced to their core concepts and do not negatively impact performance like conventional approaches that project the prototypes to real data.

The reviewers raised a diverse set of concern. The main ones are the following:
1/ Limited domain generalisation.
2/ Restricted to simple classification based tasks.
3/ High computational cost.
4/ Not convinced about the idea of using LLMs (plural) to improve prototypes.

**Reviewer Concerns:**

The authors revised the manuscript to improve the presentation and provided detailed responses to the many concerns raised by the reviewers. They clarification provided were useful. The authors acknowledged 1/ and 2/ and explained their choices. The authors acknowledged the high computational cost, but argued that the most expensive part was a one of. Finally, the authors convincingly resolved 4/ in their rebuttal.

They further specifically addressed the following concerns:
- Loss of concepts. The authors acknowledged the confusion and elaborated on the explanation.
- Usefulness of our method (for humans). The authors expanded on the discussion, addressing the concern IMO.
- Baselines. Authors provided new supporting evidence.

**Reviewer Scores:**

All reviewers voted for rejection initially. The authors did a good job at addressing their concerns and I would have expected several of them to raise their score. However, due to the many weaknesses of the original submission and the general consensus among the reviewers that the work is not meeting the acceptance bar, I do not expect that the scores would have been raised sufficiently post rebuttal to warrant acceptance.

---

### Decision · Program_Chairs · 2026-01-26

Reject